METHODS AND RESOURCES

# Phylogenomic analysis of *Wolbachia* genomes from the Darwin Tree of Life biodiversity genomics project

**Emmelien Vancaester**  *, **Mark Blaxter**

Tree of Life, Wellcome Sanger Institute, Hinxton, United Kingdom

* ev3@sanger.ac.uk

## Abstract

The Darwin Tree of Life (DToL) project aims to sequence all described terrestrial and aquatic eukaryotic species found in Britain and Ireland. Reference genome sequences are generated from single individuals for each target species. In addition to the target genome, sequenced samples often contain genetic material from microbiomes, endosymbionts, parasites, and other cobionts. *Wolbachia* endosymbiotic bacteria are found in a diversity of terrestrial arthropods and nematodes, with supergroups A and B the most common in insects. We identified and assembled 110 complete *Wolbachia* genomes from 93 host species spanning 92 families by filtering data from 368 insect species generated by the DToL project. From 15 infected species, we assembled more than one *Wolbachia* genome, including cases where individuals carried simultaneous supergroup A and B infections. Different insect orders had distinct patterns of infection, with Lepidopteran hosts mostly infected with supergroup B, while infections in Diptera and Hymenoptera were dominated by A-type *Wolbachia*. Other than these large-scale order-level associations, host and *Wolbachia* phylogenies revealed no (or very limited) cophylogeny. This points to the occurrence of frequent host switching events, including between insect orders, in the evolutionary history of the *Wolbachia* pandemic. While supergroup A and B genomes had distinct GC% and GC skew, and B genomes had a larger core gene set and tended to be longer, it was the abundance of copies of bacteriophage WO who was a strong determinant of *Wolbachia* genome size. Mining raw genome data generated for reference genome assemblies is a robust way of identifying and analysing cobiont genomes and giving greater ecological context for their hosts.

## Introduction

The natural world is a complex web of interactions between living species. These interactions can be mutualistic, commensal, pathogenic, parasitic, predatory, or inconsequential, but each individual lives alongside a rich diversity of cobionts. Most eukaryotes associate intimately with a specific microbiota and are commonly infected by a range of microbial and other pathogens. For some microbial associates, the distinction between mutualism and pathogenicity or parasitism is fuzzy. For example, *Wolbachia* (Proteobacteria; Alphaproteobacteria;

**Data Availability Statement:** The raw data for each species analysed is available under BioProject PRJEB40665 (https://www.ebi.ac.uk/ena/browser/view/PRJEB40665). Darwin Tree of Life species and data are collated in the project portal at https://

portal.darwintreeoflife.org. The Wolbachia genome assemblies are deposited in INSDC (accession numbers can be found in S2 Table) and can also be accessed on Zenodo (https://doi.org/10.5281/zenodo.7092419).

**Funding:** This research was funded by the Wellcome Trust (206194 and 218328 to MB). The funders had no role in study design, data collection and analysis, decision to publish, or preparation of the manuscript.

**Competing interests:** The authors have declared that no competing interests exist.

**Abbreviations:** ABC, ATP-binding cassette; ANI, average nucleotide identity; CI, cytoplasmic incompatibility; DToL, Darwin Tree of Life; EAM, eukaryotic association module; INSDC, International Nucleotide Sequence Database Collaboration; MAG, metagenome-assembled genome; MLST, multilocus sequence typing; SSU rRNA, small subunit ribosomal RNA.

Rickettsiales; Anaplasmataceae; Wolbachieae) are found living intracellularly in a range of terrestrial arthropods and nematodes. No free-living *Wolbachia* are known: The association is essential for their survival. In contrast, infection with *Wolbachia* can be beneficial to hosts but is not usually essential.

*Wolbachia* were first identified as mosquito endobacteria that were maternally transmitted, through the oocyte, and that induced a range of reproductive manipulations on their hosts [1,2]. The most common manipulation by *Wolbachia* is to induce cytoplasmic incompatibility (CI). Under CI, infected females are able to mate productively with all males, but uninfected females are only able to mate with uninfected males (as mating with CI-inducing *Wolbacha*-infected males results in zygotic death). This asymmetry in fitness can drive spread of the CI-inducing *Wolbachia*. Other reproductive manipulations include feminisation of genetic males [3], male killing [4], and induction of parthenogenesis in females [5]. All these manipulations promote the transmission of infected oocytes to the next host generation and thus boost the spread of *Wolbachia*. In most species that can be infected, populations are a mix of infected and infection-free individuals, and hosts can evolve to resist infection [6,7]. While *Wolbachia* are often described as reproductive parasites, association with *Wolbachia* can sometimes have beneficial effects, providing nutritional supplementation to phloem-feeding Hemiptera [8] and enhancing host immunity to viruses and *Plasmodium* parasites [9]. Indeed, the host immunity-boosting phenotype may explain the initial spread of *Wolbachia* in previously uninfected populations. In nematodes, elimination of *Wolbachia* induces host sterility, and antibiotic treatment is an effective addition to pharmacological treatment of human-infecting, *Wolbachia*-positive filarial nematodes [10].

*Wolbachia* infection of terrestrial arthropods is very common, with nearly half of all insect species predicted to be infected [11]. *Wolbachia* can be classified using molecular phylogenetic analyses into a series of supergroups [12,13]. Supergroups C, D, and J are found only in filarial nematodes; supergroups E and F are found in both nematodes and insects; and supergroups A, B, and S (and others for which full genome data are not available) are found only in arthropods. Supergroups A and B are the most common *Wolbachia* found in terrestrial insects.

Analysis of *Wolbachia* biology has been expanded by the determination of genome sequences for many isolates. The genome sequences for *Wolbachia* from over 90 host species are publicly available, and mining of host genomic raw sequence data identified a large number of additional partial genomes [14,15]. This understanding, that cobiont genomes can be assembled from the "contamination" present in the data generated for a target host, has been especially useful for the unculturable *Wolbachia*. We now have the opportunity to survey for the presence of *Wolbachia* genomes at an unprecedented scale, as the Darwin Tree of Life (DToL) project aims to sequence all described terrestrial and aquatic eukaryotic species found in Britain and Ireland [16]. This project is using high-accuracy long read and chromatin conformation long range sequencing to generate and release publicly available chromosomal genome assemblies, meeting exact standards of contiguity and completeness, for thousands of protists, fungi, plants, and animals. Several hundred terrestrial arthropod assemblies are already available (https://portal.darwintreeoflife.org). The DToL project sequences genomes from individual, wild-caught specimens of target species, and thus will also generate data for the cobiome present in each specimen at the time of sampling. For many smaller-bodied insects, the whole organism is extracted. Where *Wolbachia* disseminates widely within an organism, it is inevitable that cobiont genomes will be sequenced alongside the host genome.

Using k-mer classification tools, it is possible to efficiently and correctly separate out cobiont data from that of the host and to deliver clean host assemblies [17–19]. The cobiont data are then available for independent assembly and analysis. Here, we present a survey of the first 368 terrestrial arthropod genome datasets produced in DToL for the presence of

*Wolbachia* and assemble over 100 new *Wolbachia* genomes. We use these to explore patterns and processes in bacterial genome evolution and coevolution of *Wolbachia* with its hosts and with its own bacteriophage parasites. Lepidopteran hosts were mostly infected with super-group B, while infections in Diptera and Hymenoptera were mainly caused by A-type *Wolbachia*. However, host and *Wolbachia* phylogenies revealed no (or very limited) cophylogeny. We show that while B genomes tended to be longer compared to supergroup A, genome size in *Wolbachia* is correlated with the level of integration of its double-stranded bacteriophage WO.

## Results

### Screening a diverse set of insect genome data for *Wolbachia* infections

We screened raw genomic sequence data and primary assemblies for 368 insect species (204 Lepidoptera, 61 Diptera, 52 Hymenoptera, 24 Coleoptera, 9 Hemiptera, 5 Trichoptera, 4 Orthoptera, 3 Ephemeroptera, 3 Plecoptera, 2 Odonata, and 1 Neuroptera) generated by DToL for the presence of *Wolbachia* (S1 Table) using the small subunit ribosomal RNA (SSU rRNA) as a marker gene. *Wolbachia* SSU sequences were detected in 111 (30%) of the species. This level of infection is not reflective of total incidence, the proportion of host species susceptible to infection, as only one individual was analysed for each taxon screened. *Wolbachia* prevalence, the proportion of infected individuals in a population, and infection intensity vary between species and between populations within a species [20,21]. Therefore, the true incidence of infection within the insect biota surveyed by DToL is likely much higher. However, the measured incidence of infection is similar to previous survey-based estimates (approximately 22%) [22,23] but, as expected, is lower than estimates deploying mathematical models to account for sampling bias (40% to 50%) [11,24]. Infection incidence was lower in Coleoptera (4/24, 17%) compared to Lepidoptera (55/204, 27%), Diptera (21/61, 34%), and Hymenoptera (23/52, 44%) (Fig 1A).

Although maternal inheritance requires that *Wolbachia* are predominantly localised in the germline, tropism to somatic cell types has been shown to be highly regulated during host development [25,26]. We did not observe a bias in infection level by analysed tissue type (S1 Fig), or by gender, with an equal prevalence of infection in samples identified as female (39/138, 28%) and male (45/153, 29%) (Fig 1B). While the DToL project aims to sequence eukaryotes from across Britain and Ireland, 82% of the samples screened were sampled from the Wytham Woods Ecological Observatory, Oxfordshire (https://www.wythamwoods.ox.ac.uk/) [27]. No correlation between sampling location and infection level was detected, with 29% of all samples collected in Wytham Woods being *Wolbachia* positive, reflective of the overall incidence level (S2 Fig).

The DToL species were sequenced using PacBio Sequel II HiFi highly accurate long read platform, generating consensus raw reads of 10 to 20 kb with base level accuracy of >99% (approximately Q30 to Q40). These long, accurate reads are ideal for assembly, particularly for bacterial genomes where the information content per base is higher than in repeat-rich eukaryotes. The average sequence length of HiFi reads identified as being derived from *Wolbachia* was 12 kb, indistinguishable from host HiFi reads. We separated and assembled all *Wolbachia* reads in each positive sample and screened these assemblies to identify complete genomes. We generated 110 complete genomes, from 93 species, of which 77 were circular (S2 Table). The average completeness of these genomes, assessed using BUSCO, was 99.3%, with a mean duplication level of 0.37%. The mean genome size of the new genomes was 1.47 Mb, which is significantly larger than the average genome size of public *Wolbachia* genomes (1.32 Mb; Wilcoxon rank sum test, $p$-value = $4.576 \times 10^{-9}$) (S3 Fig). This is likely because it is

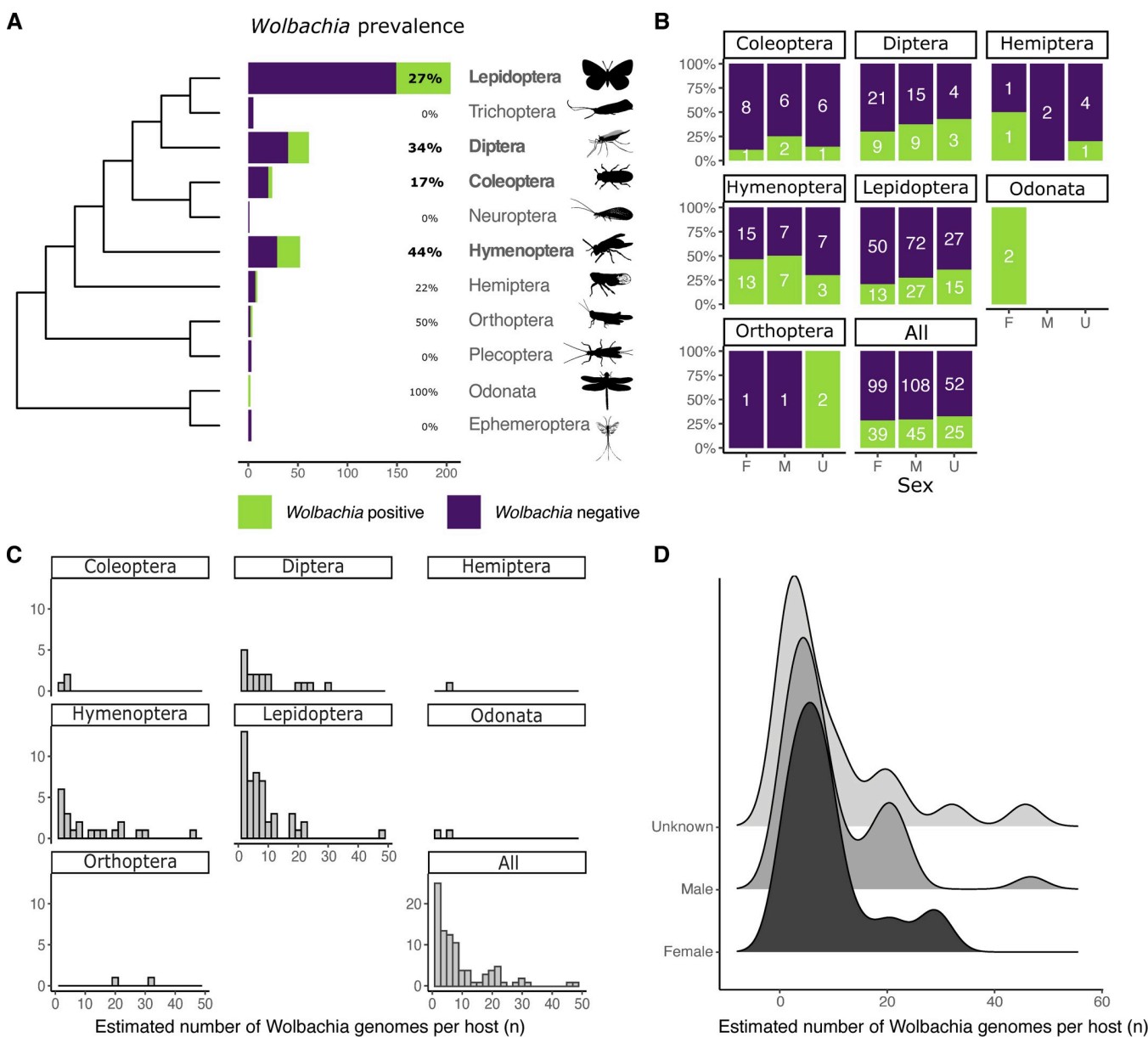

**Fig 1. Prevalence and relative abundance of *Wolbachia* in DToL insect genomes. (A, B)** Prevalence of *Wolbachia* in insect hosts, split by taxonomic order (**A**) and by sex (**B**). The cladogram of insect ordinal relationships is based on Misof and colleagues [28]. Orders with more than 10 analysed species are shown in bold. Silhouettes are from PhyloPic (http://phylopic.org/). Sex of insects was classified as F (female), M (male), or U (unknown, where not recorded on collection). The data underlying this Figure can be found in S1 Data. **(C, D)** The estimated number of *Wolbachia* genomes per copy of the host nuclear genome split by taxonomic order (**C**) and by sex (**D**). The data underlying this Figure can be found in S1 Data.

possible to assemble across repeated loci (such as integrated *Wolbachia* phage) with the long, accurate HiFi reads. The mean number of contigs generated for the 33 genomes that could not be circularised was 2.12 (ranging from 1 to 6).

The dataset includes the first complete circular *Wolbachia* genomes assembled from two insect orders, Odonata (dragonflies and damselflies) and Orthoptera (grasshoppers and crickets). Both species of dragonfly surveyed (Odonata) harboured *Wolbachia* (Fig 1A). The largest circular *Wolbachia* genome generated, 2.19 Mb, was isolated from the blue-tailed damselfly.

This is the longest complete *Wolbachia* genome yet reported (S3 Fig). Although in most samples infection by only a single *Wolbachia* strain was detected, 15 of 93 specimens (16%) were infected with at least two *Wolbachia* genomes. Within *Phalera bucephala* (Lepidoptera) and *Lasioglossum morio* (Hymenoptera), three genomes were assembled, while all other coinfections involved two strains.

Having chromosomally complete insect host genomes, as well as complete *Wolbachia*, allows for the estimation of the relative numbers of *Wolbachia* genomes per host genome. *Wolbachia* proliferation seems to be tightly controlled and a relative abundance below ten *Wolbachia* genomes per host nuclear genome was observed in most infected hosts. Particularly high abundances were observed in *Thymelicus sylvestris* and *Athalia cordata* (48 and 47 *Wolbachia* per host genome, respectively) (S2 Table) (Fig 1C). The mean relative abundance in different taxonomic orders lay between 3 and 12, except for the two crickets (Orthoptera), *Chorthippus brunneus* and *Chorthippus parallelus*, which have a 33 and 20 *Wolbachia* genome copies per host genome, respectively (Fig 1C). No significant difference was observed between relative *Wolbachia* abundance and sex of the host (Fig 1D), with both male and female having a mean between nine and ten copies.

## *Wolbachia* phylogeny suggests frequent host switching events

We selected 93 high-contiguity and high-completeness *Wolbachia* genomes from the public INSDC databases, including genomes from *Wolbachia* infecting Nematoda (13 genomes), Arachnida (4), Isopoda (1), and several orders of Hexapoda (75) (S3 Table). Adding the 110 newly assembled genomes yielded a dataset of over 200 high-quality assemblies. We annotated all protein-coding genes in those genomes using Prodigal [29] and clustered the predicted protein sets into orthologous groups using OrthoFinder2 [30]. The resulting 634 near-single copy genes were used to infer a phylogeny of *Wolbachia* (Figs 2A and S4). From this phylogeny, we assigned each genome to the previously defined *Wolbachia* supergroups [12,13]. All newly assembled *Wolbachia* genomes belonged to either supergroup A or B. While Lepidoptera were predominantly infected with supergroup B *Wolbachia* (42/53, 80%), *Wolbachia* supergroup A was most frequent in all other insect classes (46/57, 81%). It has been previously observed that supergroup B is the most common *Wolbachia* type in Lepidoptera [22,31–33]. Of the 15 species where coinfections occurred, *Endotricha flammealis*, *Phalera bucephala*, *Philonthus cognatus*, *Protocalliphora azurea*, and *Sphaerophoria taeniata* were coinfected with strains from both A and B supergroups, and the other ten coinfections were of distinct strains within the same supergroup (S2 Table).

*Wolbachia* generally do not show strict cophylogeny with their hosts [7,21]. This pattern was also observed when comparing host and *Wolbachia* phylogenies for the supergroup A and B genomes (Fig 2B). Closely related insect species may be infected by dissimilar *Wolbachia* strains, and, conversely, closely related *Wolbachia* can infect a diverse set of insects. For example, the *Wolbachia* strains infecting the hoverfly *Eupeodes latifasciatus* and four Lepidoptera (*Pararge aegeria*, *Celastrina argiolus*, *Hylaea fasciara*, and *Watsonella binaria*) (Fig 2C) share over 99% nucleotide identity. Although horizontal transmission seems to have been a dominant pattern in the evolutionary history of *Wolbachia*, the propensity of Lepidoptera to be infected by *Wolbachia* type B underlines the importance of distribution by cospeciation. Because most of our new samples came from a single site (Wytham Woods Genomic Observatory), we were also able to explore the horizontal transfer of *Wolbachia* between hosts in a local context. Wytham Woods–derived *Wolbachia* were no more likely to be related than any other *Wolbachia* subset (S5 Fig).

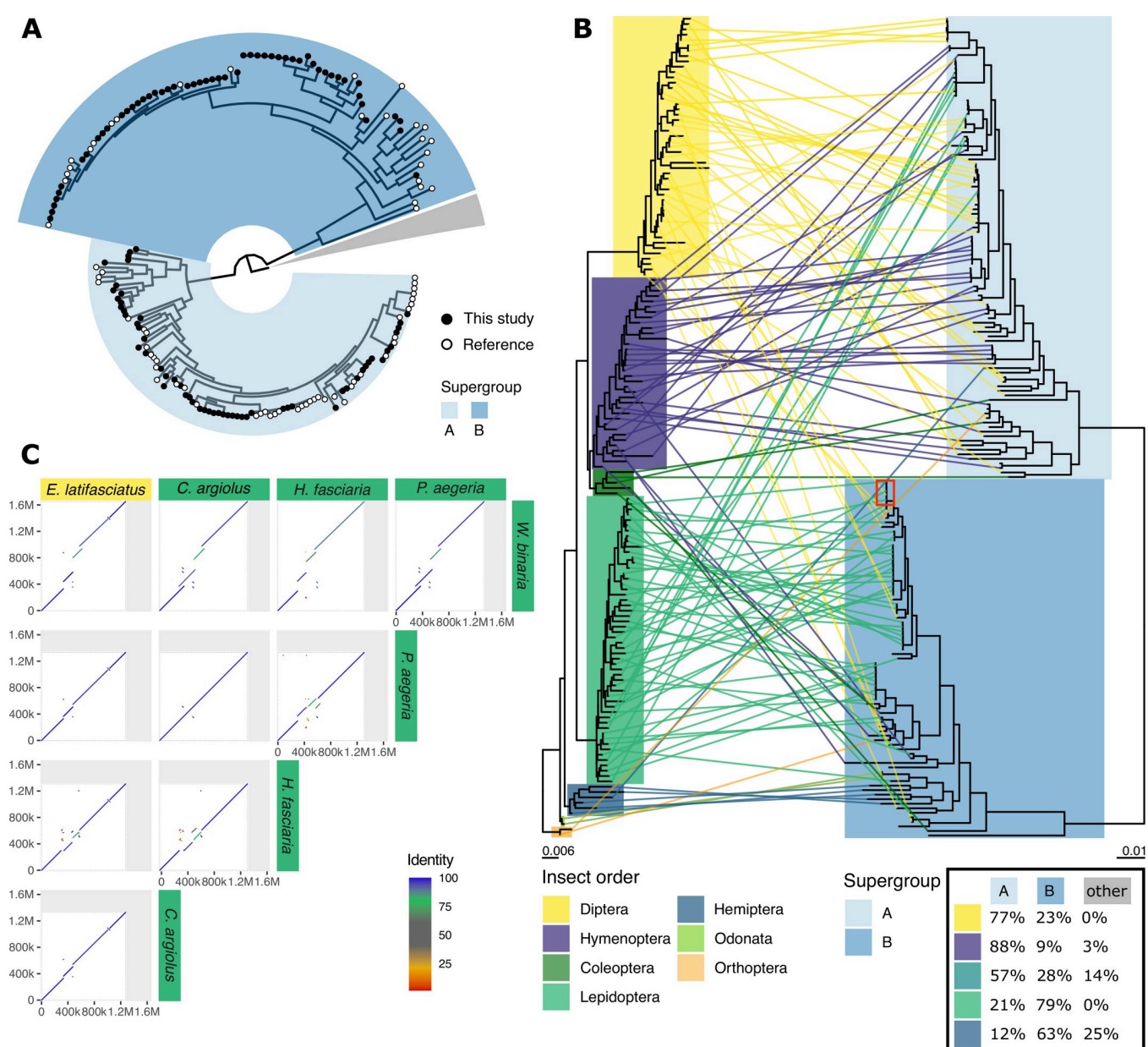

**Fig 2. *Wolbachia* DToL genomes expand known phylogeny.** (**A**) Circular phylogeny of supergroup A and B *Wolbachia*, visualised with the root placed between the A and B supergroups and the remaining supergroups (C, D, E, F, J, S; nodes collapsed as grey wedge), highlighting newly sequenced genomes (black tip labels) and genomes from public databases (white). (**B**) Incongruence between host topology (left) and supergroup A and B *Wolbachia* topology (right) is shown as a tanglegram. Overview of the supergroups infecting diverse insect orders is given in a table (inset, bottom right). A red box is drawn to point to a host switching event; see panel C. (**C**) Example of a host switching event, where the *Wolbachia* of the hoverfly *Eupeodes latifasciatus* has high nuclear sequence identity and genome colinearity to four *Wolbachia* genomes assembled from Lepidoptera.

## Intrinsic properties of *Wolbachia* distinguish supergroups

The completeness of the new genomes and, in particular, the circular assemblies achieved for 77 of them permits analyses of genome properties that are not possible with fragmented and partial genomes. All circularised genomes, including those from public databases, were rotated to start at the presumed origin of replication. The average pairwise whole-genome nucleotide identity between all *Wolbachia* genomes ranged between 77.3% and 100.0%, with at least

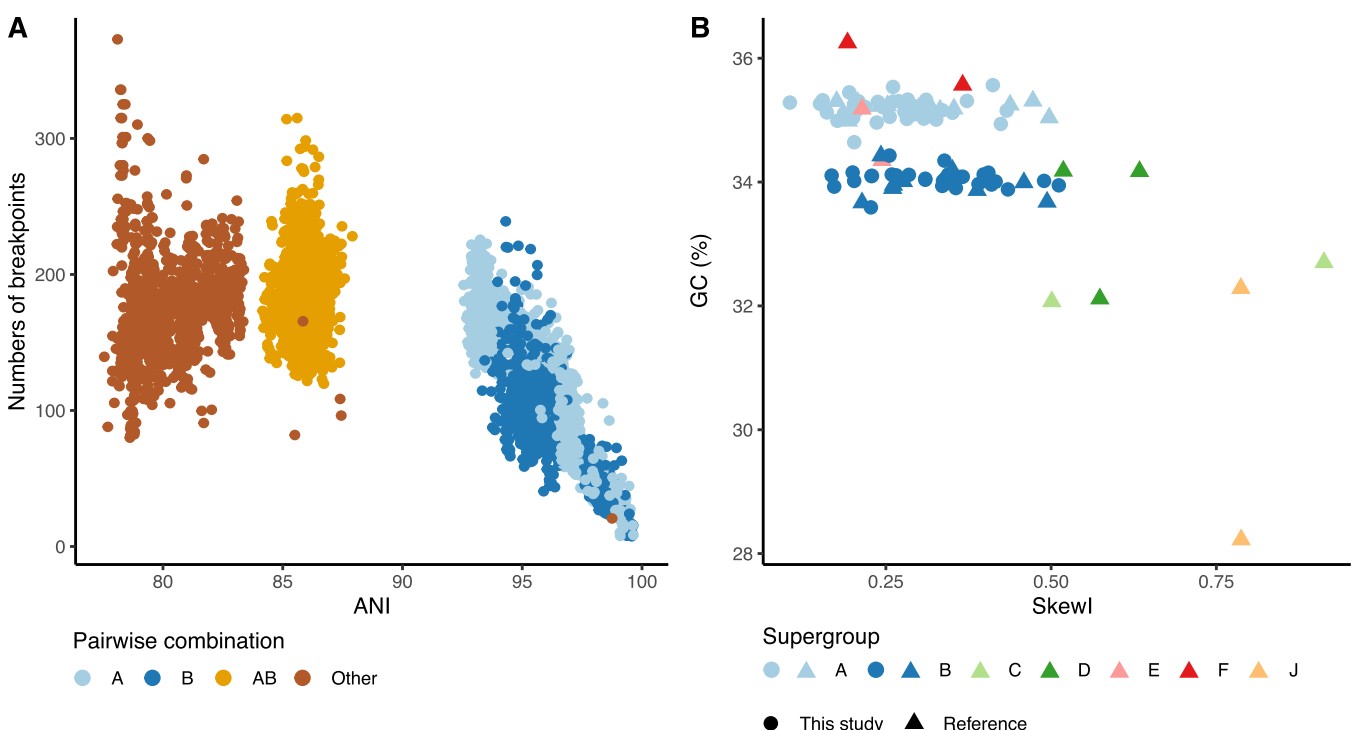

**Fig 3. Comparative genomics of *Wolbachia*. (A)** Whole-genome average nucleotide identity (ANI) plotted against the number of breakpoints in comparisons within A supergroup genomes, within B, between A and B and between other supergroup *Wolbachia*. The data underlying this Figure can be found in S1 Data. **(B)** Index of skewness compared to GC content for all circularised *Wolbachia* genomes. The data underlying this Figure can be found in S1 Data.

92.8% and 93.5% identity within supergroups A and B, respectively (Fig 3A). The number of breakpoints interrupting pairwise whole-genome alignments was counted, normalised for the total alignable length, and compared to average nucleotide identity (ANI) of the compared genomes (Fig 3A). A significant correlation was observed between nucleotide divergence and the number of breakpoints in supergroups A ($0.90$, $p < 2.2 \times 10^{-16}$, Spearman correlation) and B ($0.69$, $p < 2.2 \times 10^{-16}$, Spearman correlation) (Fig 3A). This broad range of nucleotide diversity, even within a supergroup, is indicative of the low level of conserved synteny within supergroups and the level of rearrangements occurring.

Stable bacterial genomes accumulate more guanines than cytosines on the strand in the direction of replication. This phenomenon, GC skew, arises due to differential mutation pressures on leading versus lagging strands. Genomes that have undergone frequent rearrangement are expected to have lower overall GC skew, which can be summarised across the genome as a single metric, SkewI [34]. Genomes from supergroups A and B had distinct GC contents (Fig 3B), with supergroup A having a higher mean GC (35.2%, standard deviation 0.15%), compared to B (34.0%, standard deviation 0.16%) (two-sample *t* test *p*-value $< 2.2 \times 10^{-16}$). Genomes from other supergroups had distinct GC content, often very different from A and B genomes, but as so few examples have been sequenced, general patterns are not discernible. In both A and B supergroups, SkewI values were relatively low, but genomes from Wolbachia from nematode hosts (C, D, J) had higher SkewI values (Fig 3B). A high degree of GC skew was previously reported in supergroup C *Wolbachia* strains infecting filarial nematodes [35], and these genomes also have low rearrangement levels and high gene-level synteny. In supergroups A and B, the low level of skew is associated with high levels of chromosomal rearrangement (Fig 3A).

## Conservation and diversity in gene content of *Wolbachia*

*Wolbachia*, because they are sheltered within the cells of their hosts, may be relatively isolated from other bacteria and thus have somewhat closed pan-genomes. One route to acquisition and sharing of new genes is through the *Wolbachia* phage (WO phage), which alongside the essential phage particle structural genes carry a cargo of genes that have been implicated in host manipulation. We reannotated all 203 *Wolbachia* with the same, standard gene finding toolkit, Prodigal, to normalise annotations. While this may have lost careful manual revision in previously determined gene sets, it avoids issues of data incompatibility. Gene number correlated with genome size and the average gene number in the newly assembled set of supergroup A and B *Wolbachia* was larger than in A and B genomes from the public databases (S6 Fig). Comparing all genomes, the mean number of predicted genes was larger in supergroup B (1,467) compared to A (1,385).

We used OrthoFinder with default settings to define clusters of orthologous proteins across all *Wolbachia* genomes. Each genome contained between 0 and 184 novel, strain-specific genes (average 19). These novel genes were shorter than all genes (average gene length overall was 875 nucleotides or approximately 290 amino acids, while novel genes averaged 434 nucleotides or approximately 145 amino acids). As expected, supergroups that were not well represented often contained more strain-specific genes. For example, wCfeT from supergroup E (which infects cat fleas, *Ctenocephalides felis*) uniquely encoded genes for pantothenate (panC-panG-panD-panB) [36] and thiamine (thiG-thiC) biosynthesis. Nonetheless, out of the ten genomes with most strain-specific genes, seven belonged to either supergroup A or B. These novel genes were not preferentially associated with WO phage regions (S7 Fig), but the majority (78%) had annotations that associated them with transposon and mobile element function. This suggests that much of the novelty is associated with mobile elements other than WO phage, but we note that the expansion in gene number may be due to mobile element-driven pseudogenisation. Other than clusters with one or two members, the most frequently observed cluster sizes were 203 ± 2. These clusters contained the single-copy (and near-single-copy) orthologs deployed in phylogenetic analyses (Fig 4A). Overall, the majority of the proteins encoded in the *Wolbachia* genomes were members of orthology clusters that were present in at least 95% of all strains.

The abundant sampling of supergroup A and B genomes allowed us to address and compare the sizes of the core- and pan-proteomes of these groups. The larger genome and proteome size found in supergroup B was reflected in a larger core proteome (Fig 4B), but supergroup A had a larger pan-proteome (Fig 4B). While the core proteomes differed, very few of the protein families that were part of each supergroup's core proteome were unique to that supergroup. One supergroup-restricted set of protein families was found to comprise the operon for arginine transport (ArtM, ArtQ, and ArtP and the repressor of arginine degradation ArgR) [37], which was uniquely detected and conserved in supergroup A (present in 83/103 or 80% of all *Wolbachia* A genomes). Although the periplasmic arginine-specific binding protein (ArtI or ArtJ) was not detected, the presence of this ATP-binding cassette-type (ABC) transporter suggests that these *Wolbachia* are acquiring arginine from their hosts.

The operon-producing biotin (vitamin B7) [38] was detected in seven of the 110 new genomes, all belonging to supergroup A (Fig 4C). One derived from *Icerya purchasi* (Hemiptera), and six were from Hymenoptera (two from *Lasioglossus malacharum*, which carried two strains, and single strains from three *Andrena* and a *Nomada* species). The biotin synthesis cluster has been described previously from a restricted but diverse set of supergroups, including two A genomes from additional *Nomada* bee hosts. This distribution suggests possible ecological linkage [39], as *Andrena* bees are kleptoparasitised by *Nomada* cuckoo bees and

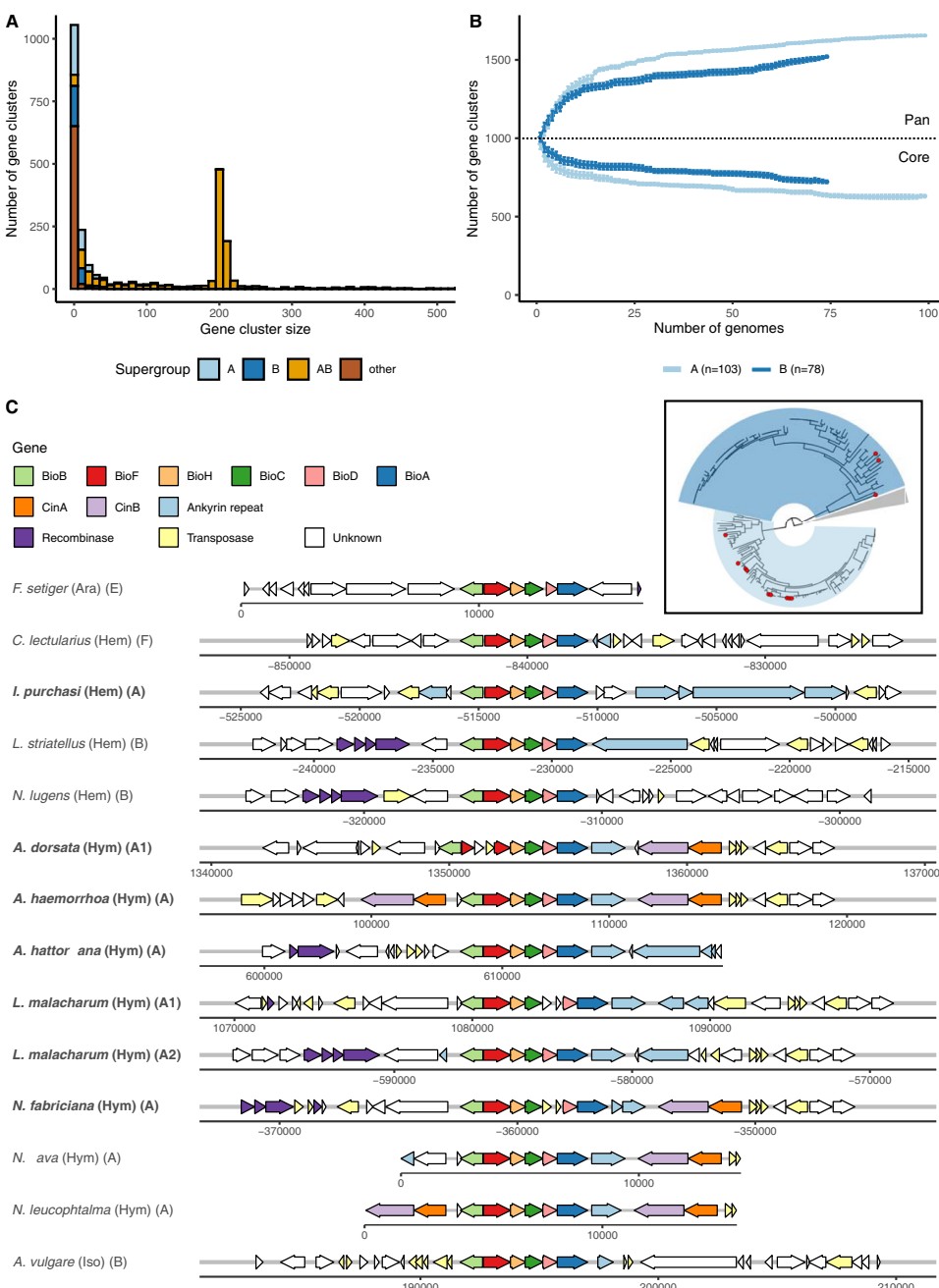

**Fig 4. Exploration of *Wolbachia* protein-coding gene diversity. (A)** Histogram of protein family size per supergroup. The data underlying this Figure can be found in S1 Data. **(B)** Rarefaction analysis of pan- and core proteomes of supergroups A and B, based on 500,000 random addition-order permutations of co-occurring orthogroups excluding novel genes. The data underlying this Figure can be found in S1 Data. **(C)** Synteny of the biotin cluster shows conserved gene order and punctuated pattern of species presence (inset, species with biotin cluster present are highlighted with red circles).

phylogenetic analyses of both the biotin gene clusters and the *Wolbachia* core proteomes show close relationships between these clusters of genomes (S8 Fig). The gene cluster is strongly conserved in physical organisation of all six necessary genes (bioA-D,F,H). In the genomic region immediately surrounding the operon, we identified recombinase and transposase

genes, as well as ankyrin repeat containing genes and toxin–antitoxin CI Cin gene pairs. In three genomes (from *Andrena dorsata*, *Nomada fabricium*, and one of the *L. malacharum* strains), the operon was independently disrupted by transposases. The region containing the biotin operon thus has the hallmarks of a "virulence island" that may be mobile between genomes and may have accrued additional genes (ankyrin, Cin) that hitchhike with the biotin operon.

## WO prophage insertions expand genome size

*Wolbachia* can itself be infected by double-stranded DNA temperate bacteriophages, WO phage, which can integrate in the genome of its host as a prophage. Four modules are necessary for construction and function of phage particles during the lytic stage: head, baseplate, tail, and fibre, and inserted and pseudogenised *WO* phage can be identified and discriminated based on the presence and completeness of these components. Regions of a *Wolbachia* genome flanked by WO phage modules are likely to form components that are transduced by the phage during infection of new cells, "cargo" loci that form the eukaryotic association module (EAM) [40,41]. All the *Wolbachia* genomes were screened for prophage regions using essential module genes from previously annotated WO insertions (S4 Table). Prophage regions were deemed putatively complete when all four modules were observed with at least 80% of genes of each module present. An abundance of putative intact and pseudogenised WO phage were identified. For example, the supergroup B *Wolbachia* from *Ischneura elegans* (the bluetail damselfly; the largest *Wolbachia* genome assembled) contained three putative intact prophage and nine WO phage fragments (Fig 5A) summing to 0.8 Mb of the genome.

The fraction of total prophage region in each genome ranged from 0% to 38%. Nematode-associated *Wolbachia* typically are not infected by WO phage [42], and no prophage regions were detected in genomes of supergroups C, D, J, and nematode-infecting F (Fig 5B). A significant correlation was found between genome size and WO prophage span in supergroups A and B (Fig 5B). This association was robust to correction for phylogenetic relatedness of the genomes (model fit increased to 0.84 and 0.87, respectively, with $p$-values $< 10^{-16}$).

## Toxins are often associated with mobile elements

We identified several potential cargo genes within intact and fragmented prophage. These included transposases and integrases associated with mobile elements, and other loci previously associated with eukaryotic manipulation, such as CI loci and ankyrin repeat containing genes, as expected from the EAM model [40,41].

*Wolbachia* produces a suite of toxins [43] that can have dramatic effects on their hosts, such as CI. The CI phenotype is caused by two adjacent genes, CifA and CifB, which function as a toxin–antitoxin pair [44,45]. Phylogenetic analysis classified most *Wolbachia* Cif gene pairs into four types (I to IV) [46]. A fifth type (V) is much more variable in structure. The toxin component can have nuclease activity (in which case the gene pair is frequently referred to as CinA-CinB), deubiquitinase (CidA-CidB), or both (CndA,CndB) [47]. All type II, III, and IV pairs have nuclease domains, while all type I have deubiquitinase and most have nuclease [46]. Three hundred and five full-length and likely functional Cif pairs were detected in 140 of the 181 (77%) supergroup A and B genomes. One Cif pair was detected in most genomes, but many had several, with seven copies in the *Wolbachia* strain infecting the holly tortrix moth (*Rhopobota naevana*). Most of the gene pairs contained a deubiquitinase domain (type I, Cid) (87) or belonged to type V (90), while the other three types occurred in roughly equal proportions (II: 39, III: 44, IV: 34) (S9 and S10 Figs). Many pairs (213/305; 70%) were located in the predicted EAM of the prophage.

Loci encoding additional toxins such as RelE/RelB and latrotoxin were identified in multiple *Wolbachia* genomes, frequently in prophage regions (175/586 [30%], 130/256 [51%] genes,

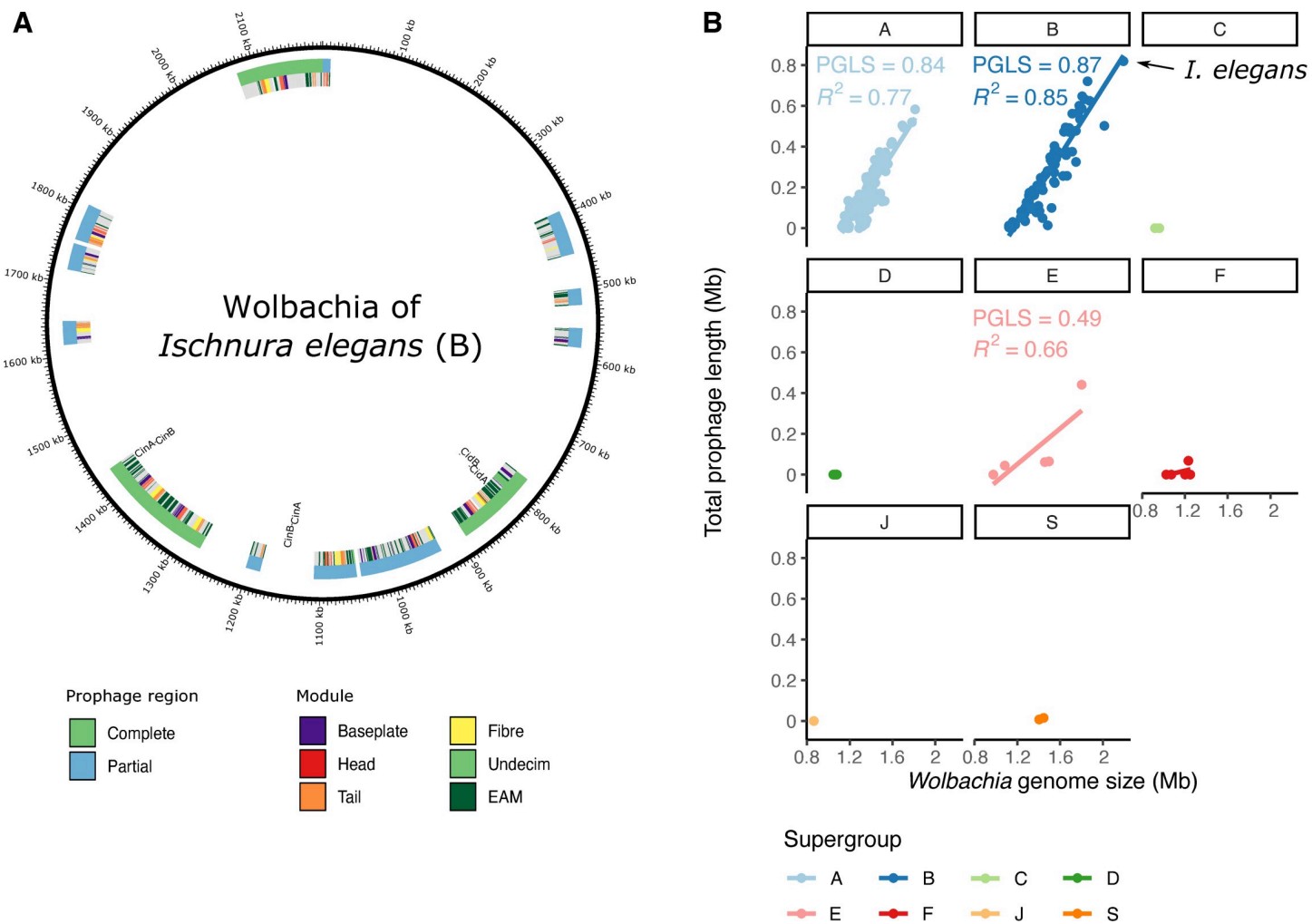

**Fig 5. WO prophage in *Wolbachia*. (A)** Annotation of the WO prophage integrated in the genome of the *Wolbachia* strain infecting *Ischnura elegans*. **(B)** *Wolbachia* genome size is strongly correlated with integrated prophage span in supergroups with WO phage association. Phylogenetic generalised least squares (PGLS) analyses were performed to assess the correlation between prophage length and genome size in a phylogenetically aware manner. The data underlying this Figure can be found in S1 Data.

respectively) (summarised in S5 Table). The Tc pore-forming toxin complex, which consists of two genes TcA (S11 Fig) and TcB-C (S12 Fig), was detected in a limited number of A and B supergroup genomes and also showed a predisposition to occur within prophage (42/69 [61%] and 19/35 [54%], respectively). Additional toxin-encoding loci had limited presence in different subgroups and were not associated with prophage regions. ParD/ParE (S13 and S14 Figs) only occurred in supergroups A, B, and E, and FIC (S15 Fig) only occurred in supergroups A, E, F, and S. The type IV toxin–antitoxin gene pair AbiEii/AbiGii-AbiEi, which protects against the spread of phage infection [48], was only detected in two genomes in supergroup E. It is noteworthy that these two genomes had very low levels of prophage-derived DNA (4.3% of their genome span).

## Discussion

Isolation of cobiont genomes, and specifically *Wolbachia* genomes, from shotgun high-throughput sequencing data has been established for many years [49]. In the field of

prokaryotic and eukaryotic microbial metagenomics, metagenome-assembled genomes (MAGs) are likely to be the only way to access many unculturable microbial genomes, even if the species they derive from are hyperabundant [50,51]. The abundance of raw sequencing data in the International Nucleotide Sequence Database Collaboration (INSDC) databases has been an attractive prospecting ground for microbial associates of eukaryotic target species. To date, most raw data available for such searches have been short reads from Illumina and other platforms. These reads are too short to partition efficiently into bins corresponding to putative distinct genomes. Preliminary assembly of such datasets is more likely to be able to separate cobionts from target genomes. These approaches have been applied to hunt for *Wolbachia* with a recent tour de force generating nearly 1,200 *Wolbachia* MAGs from publicly available data [14]. However, these MAGs suffer from the expected issues of low completeness (due to low effective coverage), fragmentation (due to coverage and sequence repeat issues), undetected contamination, and inability to distinguish coinfecting strains. Moreover, the biased nature of public data meant that these derived from only 37 different host species.

We generated 110 *Wolbachia* assemblies from 368 terrestrial arthropod HiFi datasets, and 77 of these were fully circular genome assemblies. The genomes were uniformly of high completeness (S3 Fig). Due to the high intrinsic base quality of HiFi reads (Q30 to Q40; from one error in 1,000 to one error in 10,000), we were able to distinguish insertions of *Wolbachia* DNA into the host genome from true components of the *Wolbachia* genome and to independently assemble even closely related strains with confidence. As we were screening raw data from a biodiversity genomics programme that aims to sample a wide phylogenetic diversity of hosts, the new *Wolbachia* genomes presented here more than double the number of different host species from which *Wolbachia* genomes have been assembled. The assembled genomes include the first complete representatives isolated from Odonata (damselflies) and Orthoptera (crickets). In 16 additional datasets, we identified likely *Wolbachia* content but were not able to produce credible genome assemblies (see S1 Data and S2 Table). This was usually because the *Wolbachia* sequence was present in very low effective coverage (approximately threefold), but in some samples, no credible assembly was generated despite high coverage. These datasets may contain multiple recombining strains or contain large insertions in the host genome and deserve further exploration.

The distribution of *Wolbachia* in insect hosts is a function of the balance between retention through cospeciation (vertical transmission of *Wolbachia* to daughters of the host species), acquisition through horizontal transmission (where strains move between host species), and events of loss. Transmission among insect hosts was the dominant pattern underpinning *Wolbachia* distribution. We note that previous work has suggested that horizontal transmission rather than cospeciation may even explain the presence of closely related *Wolbachia* infecting closely related taxa. For example, genomic divergence between closely related *Wolbachia* in sister *Drosophila* species was too low to be the product of independent evolution since the last common ancestor of the flies [52,53]. However, we identified two features of the distribution, one local and one general, which are of note. Lepidoptera were more likely to be infected with supergroup B *Wolbachia* than A, and Hymenoptera, Diptera, and Coleoptera were more likely to be infected with supergroup A strains. Multilocus sequence typing (MLST) has previously shown that supergroup B is the most common *Wolbachia* type in Lepidoptera [22,31–33]. This suggests some nonexclusive specialisation of *Wolbachia* on their hosts, which may be driven by the interaction of *Wolbachia* and host genetics and/or a distinct set of ecological transmission routes in each insect group. Many of our genomes derived from insects were collected at one site, the Wytham Woods Genomic Observatory (S2 Fig), but this subset was no more closely related than other genomes from widely separated sites (S5 Fig). It is likely that the mobility of hosts, including through seasonal migration, means that sampling from one

geographical site is a valid approximation of more global sampling. Close ecological association between host species may promote sharing of *Wolbachia* isolates and localised genetic exchange, for example, within predator–prey systems. The close similarity of *Wolbachia* genomes from *Andrena* solitary bees and their *Nomada* cuckoo bee kleptoparasites (Fig 4C, inset) and the shared occurrence of the biotin synthesis operon (Fig 4C) may be a case of transmission within an ecological network. The presence of the biotin operon in *Wolbachia* of insects that largely or solely feed on low-protein plant fluids (nectar or phloem) suggests that *Wolbachia* may be offering nutritional support to their hosts [54] and thus that this cluster of genomes may have been positively selected for their mutualist tendencies.

*Wolbachia* can promote reproductive success of their female hosts [1,2], and thus their own Darwinian fitness, through reproductive manipulations such as CI. The loci underpinning CI are a diverse set of toxin–antitoxin gene pairs. Our survey of *Wolbachia* identified many additional CI gene pairs, mainly of the I Cid type and mostly associated with WO phage. Many genomes had more than one toxin–antitoxin pair, and some individual hosts were infected with multiple *Wolbachia* strains carrying different CI gene pairs. These CI genes likely mediate conflict between *Wolbachia* strains and the ecosystem of toxin deposition and rescue in individual zygotes must be complex [46,55,56]. Interestingly, we identified CI gene pairs next to 5 of the 14 biotin synthesis operons, suggesting that the mobile elements that transduce this presumably mutualist physiology are also engaged in CI conflict.

One striking feature of the genomes assembled from the HiFi reads was that their average span was approximately 10% greater than the average size of previously assembled *Wolbachia* genomes. As we also observed a correlation between content of WO phage in the genome and genome size (Fig 5B), we speculate that the lower average size of previous assemblies may be because the presence of near-identical segments of phage and other mobile elements led to collapse of repeats and artificial underestimation of true genome size. This underestimation of genome size may also have biased understanding of WO phage diversity and of the diversity of genes that can be transduced by the phage. WO phage carry genes necessary for production of phage particles and cargo genes that have been hypothesised to form an EAM [40,41]. The increased genome size and increased resolution of WO phage copies might also mean increased gene content and diversity and an increased set of common EAM loci. We estimated the pan-proteome of A and B supergroup strains and found that the supergroup A had a higher pan-proteome but a smaller core proteome than supergroup B. Coupled with the observation of host-association bias between these supergroups, and other major genomic features such as GC proportion, this suggests that these divergent groups have followed very distinct evolutionary trajectories, despite evidence for transduction of loci between supergroups, and perhaps have evolved distinct physiologies and host-manipulation or host-cooperation strategies. We note that the ANI between A and B supergroup strains, and between strains from all supergroups, is relatively low (within-supergroup identity >93%, between-supergroup identity <88%). This pattern of significant phylogenetic separation between supergroups suggests, as others have noted, that these supergroups have the features expected of bacterial species [37].

The DToL project [16] is one of a growing constellation of biodiversity genomics initiatives worldwide that, under the banner of the Earth BioGenome Project [57], intend to "sequence life for the future of life" (https://www.earthbiogenome.org). These projects, based around ecological, regional, or taxonomic lists of target species, will lay the foundations for biological research, bioindustry, and conservation for the next decades. While their focus is to generate reference genomes for eukaryotic species, these projects will also yield critical resources for the study of the microbial cobionts—mutualists, pathogens, parasites, and commensals—which live on and in eukaryotic organisms. Our understanding of *Wolbachia* and other common endosymbionts will thrive on a rich harvest of cobiont genomes from the tens to hundreds of

thousands of host genomes that will be generated in the next decade. The assembly of 110 high-quality *Wolbachia* genomes shows the power of the long read data now being generated and the analytic approach that allowed these low complexity metagenomes to be effectively separated into their constituent parts. Analysis of these genomes revealed a propensity to infect different insect orders among supergroups, while simultaneously pinpointing to several host switching events during the course of the *Wolbachia* pandemic. Moreover, we observed that genome size in *Wolbachia* is correlated with the abundance of copies of bacteriophage WO.

## Methods

### Detection and assembly of *Wolbachia* genomes from DToL species data

DToL raw data are generated from whole or partial single specimens and thus contain sequence from any cobionts in or on the specimen at the time of sampling. We screened data for 368 insect genomes generated by the DToL project [16] for the presence of the intracellular endosymbiont *Wolbachia* (S1 Table) using a marker gene scan approach by searching for the SSU rRNA locus. The prokaryotic 16S rRNA alignment from RFAM (RF00177) [58] was transformed into a HMMER profile, and the profile was used to screen contigs with nhmmscan [59]. We defined a positive match as having an e-value $<10^{-150}$ or an aligned length of $>1,000$ nucleotides. Putative positive regions were extracted from the sequences and compared to the SILVA SSU database (version 138.1) [60] using sina [61]. Matches were filtered to retain only those with $>90\%$ identity. Taxonomic classification of each positive was determined via a consensus rule of 80% of the top 20 best hits, using both the NCBI [62] and SILVA [63] taxonomies.

For *Wolbachia*-positive samples, all PacBio HiFi reads were analysed using kraken2 [64] with a custom database consisting of a genome from a species closely related to the host, all RefSeq genomes of Anaplasmataceae, and reference genomes of additionally detected cobionts downloaded using NCBI datasets and masked using dustmasker [65]. Horizontal transfer of fragments of endosymbiont and organellar DNA to the nuclear genome is a common phenomenon. To avoid inadvertently classifying nuclear *Wolbachia* insertions (NUWTs) as deriving from an independent bacterial replicon, *Wolbachia* reads identified by kraken2 were mapped to the insect genome assembly, and only contigs fully covered by these reads were retained. The *Wolbachia* reads were also independently reassembled using several assembly tools: flye (version 2.9) (flye—pacbio-hifi {reads} -o {dir} -t {threads}—asm-coverage 50—genome-size 1.6m —scaffold) [66], hifiasm (version 0.14) (hifiasm -o {prefix} -t {threads} {reads} -D 10 -l 1 -s 0.999) [67], and hifiasm-meta (version 0.1-r022) (hifiasm_meta -o {prefix} -t {threads} {reads} -l 1) [68]. The several assemblies generated for each sample were ranked based on their completeness using BUSCO version 5.2.2 [69] and the Rickettsiales_odb10 dataset, alignment to reference genomes using nucmer (version 4.0.0) [70], evenness of coverage, and circularity. The best (most complete, single-contig circular preferred) assembly per sample was chosen. For samples where 10X Genomics Chromium data were available, polishing was performed using FreeBayes-called variants [71] from 10X short reads aligned with LongRanger. The host origin, span, and completeness of all *Wolbachia* detected are presented in S2 Table.

### Collation of *Wolbachia* genome dataset, gene prediction, and orthology inference

All available *Wolbachia* genomes were downloaded from NCBI GenBank on 01/02/2022 and supplemented with assemblies generated from short-read insect datasets by Scholz and

colleagues [14]. This dataset contained replicate genomes for very closely related *Wolbachia* from the same host, and many fragmented and partial assemblies. Only the most contiguous assembly per host species was retained. These genomes were renamed using the schema "R_Xyz_GenSpec_§", where Xyz is the first three letters of the insect order of the host, Gen-Spec is an abbreviation derived from the generic and specific epithets of the host, and § indicates the supergroup. Retained assemblies were assessed for the presence of contamination by performing a contig analysis by kraken2 using a database of only circular *Wolbachia* genomes. A list of all removed contigs can be found in S3 Table. Furthermore, we only included database-sourced *Wolbachia* genomes with at least 90% BUSCO completeness [69] and at most 3% duplication with the Rickettsiales_odb10 dataset (S3 Table). The exception to this filtering was the inclusion of genomes belonging to the most divergent supergroup S.

All of the publicly available and newly assembled genomes were annotated using Prodigal (version 2.6.3) [29]. Protein families were inferred using OrthoFinder (version 2.4.0) [30]. We identified 624 protein families, which were single-copy in more than 95% of all *Wolbachia* genomes. These were individually aligned using mafft in automatic mode (version 7.490) [72]. Individual maximum likelihood gene trees were calculated using iqtree (version 2.1.4) (iqtree -s {alignment} -nt {threads}) [73], and coalescence of these gene trees was determined using ASTRAL (version 5.7.4) [74]. The individual alignments were trimmed using trimAl (version 1.4) [75] and concatenated to form a supermatrix. This was used to infer a maximum likelihood phylogeny with iqtree using 1,000 ultrafast bootstrap approximation iterations (version 2.1.4) (iqtree -s {supermatrix} -m LG+G4 -bb 1000 -nt {threads}) [73]. The insect topology was subsampled from Chesters [76]. Incongruence in topology between the insect host and *Wolbachia*, host phylogeny was determined with ggtree in R [77].

## Intrinsic genomic properties

All circular genomes were rotated to start with HemE (OG0000716) on the positive strand, as this gene is located next to the origin of replication [78]. All pairwise alignments were calculated using nucmer (version 4.0.0) [70], and breakpoints were inferred and adjusted for the aligned coverage. Whole-genome average nucleotide diversity was calculated using FastANI (version 1.33) [79]. GC and GC skew index values were calculated for all genomes using SkewIT [34].

## Gene content analysis

To functionally annotate predicted genes, both Prokka (version 1.14.6) [80] and InterProScan (version 5.54–87.0) [81] were run. The synteny plot of the biotin locus was created using gggenes [82]. All six genes that make up the biotin locus (BioA-D, BioF, BioH) were individually aligned with mafft in automatic mode (version 7.490) [72] and transformed into a concatenated nucleotide alignment. A phylogenetic tree was built using the model GTR+F+G4 in iqtree (version 2.1.4) [73]. Genes responsible for CI were identified by a BLAST search [83] using the following genes as queries: CidA: WP_010962721.1, WP_182158704.1, WP_012673228.1, WP_006014162.1, CAQ54402.1, NZ_MUIX01000001.1_1324, OAM06111.1; CifB: WP_010962722.1, WP_182158703.1, WP_012673227.1, WP_006014164.1, CAQ54403.1, NZ_MUIX01000001.1_1323, OAM06112.1. Moreover, additional CifB type V genes were added as reference genes (Diachasma_alloeum_pair1, Diploeciton_nevermanni_pair5, wBor_-pair2, wStri_pair1, wStri_pair2 and wTri-2_pair1). Only pairs of identified neighbouring genes (e-value $1 \times 10^{-30}$, coverage 80% to 120%) were retained. Both CifA and CifB were aligned using mafft in automatic mode (version 7.490) [72], followed by maximum likelihood estimation using iqtree (version 2.1.4) (iqtree -s {alignment} -nt {threads} -bb 10000).

## WO prophage analysis

A list of known prophage sequences was generated based on annotated regions described in the literature [41,44,84] (S4 Table) for a set of genomes (R_Dip_DroSim_A, R_Hym_Nas-Vit_A, R_Dip_DroAna_A, R_Dip_HaeIrr_A, R_Hym_CerSol_A, and R_Hym_WiePum_A) and linked to their respective gene families. Each *Wolbachia* genome was screened for continuous stretches of linked prophage genes with at most five other genes in-between, and these were annotated as prophage regions if they contained at least one gene from one of the four core phage modules (head, baseplate, tail, and fibre). This permitted detection of novel prophage-associated genes. Regions that contained at least 5 of 6 head, 7 of 8 baseplate, 5 of 6 fibre, and 5 of 6 tail module genes were deemed putatively complete. Genomic maps of prophage integration were created with circos [85]. Phylogenetic generalised least squares analyses were performed to assess the correlation between prophage length and genome size using the ape R package [86], using a Brownian model of evolution and the phylogenetic tree in Fig 2A. R squared values were calculated using the package rr2 [87].

## Supporting information

**S1 Data. Supplementary Data.**
(XLSX)

**S1 Table. DToL screened genomes.**
(PDF)

**S2 Table. Overview detected *Wolbachia* genomes.**
(PDF)

**S3 Table. *Wolbachia* reference genomes.**
(PDF)

**S4 Table. Prophage modules.**
(PDF)

**S5 Table. Summary toxin genes.**
(PDF)

**S1 Fig. Selected tissue and incidence of *Wolbachia*.** Selected tissue and incidence of *Wolbachia* presence (green) and absence (purple) of DToL samples.
(PDF)

**S2 Fig. Sampling locations and incidence of *Wolbachia*.** Sampling locations and incidence of *Wolbachia* presence (green) and absence (purple) of DToL samples from Britain and Ireland. The map was drawn using the maps library (version 3.4.0) in R, which imports data from the public domain (Natural Earth project) (https://www.naturalearthdata.com/downloads/50m-physical-vectors/). The size of the pie charts reflects the number of collected samples per location. Most samples came from Wytham Woods Genomic Observatory near Oxford. The data underlying this Figure can be found in S1 Data.
(PDF)

**S3 Fig. Contiguity and genome size distribution of *Wolbachia*.** (**A**, **B**) Contiguity and genome size distribution of *Wolbachia* genomes assembled in this study (black) vs. reference genomes from other projects available in NCBI (grey). The data underlying this Figure can be found in S1 Data. (**C**) Genome size distribution of *Wolbachia*. Supergroups A (above) and B (below), in this study (black) and reference genomes from other projects available in NCBI

(grey) were compared by Wilcoxon rank sum test. The data underlying this Figure can be found in S1 Data.
(PDF)

**S4 Fig. Phylogeny of supergroup A and B *Wolbachia*.** Phylogeny of supergroup A and B *Wolbachia*, visualised with the root placed between the A and B supergroups and the remaining supergroups (C, D, E, F, J, S; nodes collapsed as grey wedge), highlighting nodes with bootstrap value higher than 80 with a black label.
(PDF)

**S5 Fig. Average nucleotide identity between Wytham Woods specimens.** Distribution of average nucleotide identity (ANI) between pairs of *Wolbachia* genomes if specimens were both sampled from Wytham Woods (upper panel) or any other locality (lower panel). Distributions are separated by the classification of the two genomes, i.e., both belonging to supergroup A, both belonging to supergroup B, comparisons of A with B, or comparisons between other supergroups. The data underlying this Figure can be found in S1 Data.
(PDF)

**S6 Fig. Predicted proteome size in *Wolbachia*.** Number of predicted protein-coding genes for *Wolbachia* supergroups A (above) and B (below), in this study (black) and reference genomes from other projects available in NCBI (grey) were compared by Wilcoxon rank sum test. The data underlying this Figure can be found in S1 Data.
(PDF)

**S7 Fig. Strain-specific proteins are not generally associated with WO phage.** Percentage of protein-coding genes present in WO prophage regions versus percentage of strain-specific protein-coding genes in those regions of *Wolbachia* genomes with at least 10 strain-specific genes. Size of points is reflective of the total number of strain-specific genes. Linear regression line with confidence interval is displayed. The data underlying this Figure can be found in S1 Data.
(PDF)

**S8 Fig. Comparison of the phylogenies of biotin synthesis clusters and the *Wolbachia* strains that contain them.** Comparison between phylogenies of *Wolbachia* genomes containing the biotin locus, based on tree in Fig 2A (left) and a phylogeny inferred from the six nucleotide genes constituting the biotin synthesis operon (BioA-D, BioF, BioH) (right). Internal nodes with bootstrap support higher than 80 are highlighted with black circles.
(PDF)

**S9 Fig. Phylogenetic representation of detected CifA genes.** Phylogeny of CifA toxin genes, highlighting nodes with a bootstrap value higher than 80 with a circle.
(PDF)

**S10 Fig. Phylogenetic representation of detected CifB genes.** Phylogeny of CifB toxin genes, highlighting nodes with a bootstrap value higher than 80 with a circle.
(PDF)

**S11 Fig. Phylogenetic representation of detected TcA genes.** Phylogeny of TcA toxin genes, highlighting nodes with a bootstrap value higher than 80 with a circle.
(PDF)

**S12 Fig. Phylogenetic representation of detected TcB-C genes.** Phylogeny of TcB-C toxin genes, highlighting nodes with a bootstrap value higher than 80 with a circle.
(PDF)

**S13 Fig. Phylogenetic representation of detected ParD genes.** Phylogeny of ParD toxin genes, highlighting nodes with a bootstrap value higher than 80 with a circle.
(PDF)

**S14 Fig. Phylogenetic representation of detected ParE genes.** Phylogeny of ParE toxin genes, highlighting nodes with a bootstrap value higher than 80 with a circle.
(PDF)

**S15 Fig. Phylogenetic representation of detected FIC genes.** Phylogeny of FIC toxin genes, highlighting nodes with a bootstrap value higher than 80 with a circle.
(PDF)

## Acknowledgments

We thank our many colleagues in the Darwin Tree of Life project—from field collectors to data curators—for the production of the raw data we analysed. We also thank Tree of Life colleagues, especially Claudia Weber, Charlotte Wright, and Ellen Cameron for fruitful discussions and Andrew Varley, James Torrance, and Shane McCarthy for help with sequence deposition.

## Author Contributions

**Conceptualization:** Emmelien Vancaester, Mark Blaxter.

**Data curation:** Emmelien Vancaester, Mark Blaxter.

**Formal analysis:** Emmelien Vancaester.

**Funding acquisition:** Mark Blaxter.

**Investigation:** Emmelien Vancaester, Mark Blaxter.

**Methodology:** Emmelien Vancaester.

**Project administration:** Mark Blaxter.

**Software:** Emmelien Vancaester.

**Supervision:** Mark Blaxter.

**Validation:** Emmelien Vancaester, Mark Blaxter.

**Visualization:** Emmelien Vancaester.

**Writing – original draft:** Emmelien Vancaester, Mark Blaxter.

**Writing – review & editing:** Emmelien Vancaester, Mark Blaxter.

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
