## [Editor Report · Decision Letter 0]

13 Oct 2022

Dear Dr Vancaester, 

Thank you for submitting your manuscript entitled "An endosymbiont harvest: Phylogenomic analysis of Wolbachia genomes from the Darwin Tree of Life biodiversity genomics project." for consideration as a Research Article by PLOS Biology.

Your manuscript has now been evaluated by the PLOS Biology editorial staff, as well as by an academic editor with relevant expertise, and I am writing to let you know that we would like to send your submission out for external peer review. I should warn you that the Academic Editor was slightly unsure as to whether your study was better suited to PLOS Biology or PLOS Genetics, so we will be asking the reviewers whether the advance is sufficient for PLOS Biology.

Once your full submission is complete, your paper will undergo a series of checks in preparation for peer review. After your manuscript has passed the checks it will be sent out for review. To provide the metadata for your submission, please Login to Editorial Manager (https://www.editorialmanager.com/pbiology) within two working days, i.e. by Oct 17 2022 11:59PM.

Kind regards,

Roli Roberts

Roland Roberts, PhD

Senior Editor

PLOS Biology

rroberts@plos.org

---

## [Decision Letter · Decision Letter 1]

18 Nov 2022

Dear Dr Vancaester,

Thank you for your patience while your manuscript "An endosymbiont harvest: Phylogenomic analysis of Wolbachia genomes from the Darwin Tree of Life biodiversity genomics project." was peer-reviewed at PLOS Biology. It has now been evaluated by the PLOS Biology editors, an Academic Editor with relevant expertise, and by three independent reviewers. 

Based on the reviews, we are likely to accept this manuscript for publication, provided you satisfactorily address the points raised by the reviewers. Please also make sure to address the following data and other policy-related requests.

IMPORTANT: Please address the following:

a) We try to avoid punctuation in titles; please could you N-terminally truncate yours to "Phylogenomic analysis of Wolbachia genomes from the Darwin Tree of Life biodiversity genomics project"?

b) Given the concerns raised by reviewer #3 about the limited novel biological insights (these were shared by the staff editors and Academic Editor before review), and in recognition that your study is nonetheless potentially of significant importance to our readership, please change the article type to "Methods and Resources" when you re-submit. No re-formatting is required.

c) Please attend to the other requests from the reviewers.

d) We note that reviewer #2 suggests some additional analyses that would certainly add value and interest to your study (e.g. wmk and toxin genes); we leave it to you to decide whether to include these.

e) Please provide a blurb, according to the instructions in the submission form.

f) Please address my Data Policy requests below; specifically, we need you to supply the numerical values underlying Figs 1ACD, 2ABC, 3AB, 4AB, 5B, S1, S2ABC, S3, S4, S5, S6, S7, either as a supplementary data file or as a permanent DOI’d deposition.

g) Please cite the location of the data clearly in all relevant main and supplementary Figure legends, e.g. “The data underlying this Figure can be found in S1 Data” or “The data underlying this Figure can be found in https://doi.org/XXXX”

We expect to receive your revised manuscript within three weeks. 

*Published Peer Review History*

*Press*

Sincerely,

Roli Roberts

Roland Roberts, PhD

Senior Editor,

rroberts@plos.org,

PLOS Biology

DATA POLICY:

Regardless of the method selected, please ensure that you provide the individual numerical values that underlie the summary data displayed in the following figure panels as they are essential for readers to assess your analysis and to reproduce it: Figs 1ACD, 2ABC, 3AB, 4AB, 5B, S1, S2ABC, S3, S4, S5, S6, S7. NOTE: the numerical data provided should include all replicates AND the way in which the plotted mean and errors were derived (it should not present only the mean/average values).

We require the original, uncropped and minimally adjusted images supporting all blot and gel results reported in an article's figures or Supporting Information files. We will require these files before a manuscript can be accepted so please prepare and upload them now. Please carefully read our guidelines for how to prepare and upload this data: https://journals.plos.org/plosbiology/s/figures#loc-blot-and-gel-reporting-requirements

DATA NOT SHOWN?

REVIEWERS' COMMENTS:

Reviewer #1:

[identifies himself as Julien Martinez]

Studying the biology of intracellular symbionts like Wolbachia is challenging since they are often not amenable to genetic manipulation. Comparative genomics has been an important tool in identifying the genetic basis of symbiont-induced phenotypes such as reproductive manipulations and in understanding the major evolutionary forces that shape symbiont genomes. To date, a significant proportion of Wolbachia assemblies publicly available are fragmented as they were often generated using short read sequencing technologies which prevents the assembly of large repeat regions that are abundant in Wolbachia. This has been a limitation for the studying the evolution of symbiont gene content and genome architecture. Moreover, available genomes are biased towards certain Wolbachia-carrying host taxa and model organisms, and it is unclear how one can generalize observations such as the incidence of certain Wolbachia clades in nature.

The present study by Vancaester & Blaxter generated a large number of high quality and complete to near-complete Wolbachia genomes isolated from a wide diversity of host taxa which is an important step to overcome these challenges. By analyzing these new genomes along with publicly available assemblies, the authors provide general insights into Wolbachia evolutionary genomics such as the role of prophage sequences in genome size variation. Another important finding is that the new genomes presented here are bigger on average than previous Wolbachia assemblies which highlights the need for using high-accuracy long-read sequencing technologies.

This new set of genomes will be of high interest to the scientific community and will undoubtedly facilitate more in-depth analyses of Wolbachia evolutionary genomics. I have some comments/suggestions below that I hope the authors will find useful. Great work!

1) Line 73: the authors should also refer to the Pascar & Chandler 2018 study (PMID: 30202647) who generated many Wolbachia draft genomes using a similar approach.

2) Line 85-86: Does that mean that somatic tissues were selected for sequencing and ovaries generally discarded? Maybe specify. If that is the case, it would be useful to make it clear that some Wolbachia infections in this collection of insects may have been missed since Wolbachia infections can vary greatly in their tissue distribution, some being restricted to germline tissues (e.g. see Strunov et al 2022, PMID: 35357208).

3) Line 107-109: this is also in line with the more recent Weinert et al. 2015 study which accounted for sampling bias (PMID: 25904667).

4) Lines 114-120: it might be useful to clearly define the difference between prevalence (proportion of infected individuals in a population/species) and incidence (proportion of host species where the symbiont is present in a given host clade).

5) Line 146-148: there is evidence in the literature that Wolbachia titer/abundance within hosts can be controlled both by the host and the bacterial genomes. However, for most associations, the genetic determinants of Wolbachia proliferation have not been characterized. Therefore the statement "Most infected hosts tightly control Wolbachia proliferation" is inaccurate/not supported since looking at Wolbachia titers alone does not tell us whether it is controlled by the host, the symbiont or both.

6) Line 149: 48 Wolbachia per host "genome".

7) Line 259-262: I suspect that a lot of these "novel genes" might be pieces of pseudogenes. The fact that they were found to be much shorter on average and mostly annotated as transposon/mobile elements is not surprising since transposase and reverse transcriptase genes are abundant in Wolbachia genomes and are often highly degraded. On top of this, these mobile elements often insert themselves within and disrupt other genes. As I understand, the genome annotations were not manually curated to reannotate pseudogenes and I wonder how much of these "novel genes" are simply degraded/split/truncated copies of genes that are present in full length in other genomes but that OrthoFinder failed to place in the correct orthogroup, instead clustering them into a one/two member orthogroup. Could the authors elaborate on this and mention in the manuscript whether they think this is a limitation for defining what a novel gene is? I guess this is to be kept in mind when estimating the size of the core/pan-genomes and looking at variation between Wolbachia supergroups (some supergroups could have more degraded genomes for example which could artificially inflate the number of orthogroups detected).

8) Lines 314-315: can the authors explain their rationale for using the 80% threshold to define a prophage region as complete and what they mean by "complete"? If a prophage region carries >80% of genes of each phage module but is lacking an essential structural phage gene preventing it to produce phage particles, should it be called complete? Also, in line 476, defining a phage copy as "active" would mean that there is some evidence the phage is replicating and/or producing phage particles. I was wondering if the authors have any indication on which prophage region might be active, based on variation in sequencing depth. If not, I would probably avoid calling them active.

9) Lines 397-399: the balance also depends on loss of infections through time (not only gains).

10) Lines 345-351: from the method section, it seems that the authors used a coverage threshold for the detection of cif genes that should miss typical type V cif genes. If I understand correctly, representative cif genes from Type I (cidA/B) and Type IV (cinA/B) were used as queries and only hits that had 80-120% coverage were retained. However, type V cifB genes are typically much longer than type I-IV (4-5x longer) due to the presence of additional domains such as ankyrin repeat and a C-terminal latrotoxin domain (some type V cifB genes are shorter due to premature stop codons indicating pseudogenization or streamlining processes, however, the truncated ankyrin/latrotoxin domains are often found downstream of the disrupted genes). Therefore, my guess is that more full-length type V cif genes are present in this set of new genomes than reported in line 350 (50 type V homologues). For that reason, I also wonder how many of the latrotoxin domain-containing proteins reported in line 352 are in fact full copy or the 3'-end of truncated type V cifB proteins. I would suggest that the authors use representative type V cif genes in addition to type I-IV as queries or instead mention that they might be missing a lot of type V genes in their analysis. Another solution would be to remove the 120% maximum coverage threshold to include the longer cifB genes.

11) Table S3: The reference genome accession GCF_001931755.2 was isolated from the Springtail Folsomia candida (Collembolla), not from a coleopteran host as indicated in Table S3.

Reviewer #2:

Wolbachia is the most successful host-associated microbe on the planet, estimated to infect ~40% of all terrestrial arthropod species. These symbionts are transmitted primarily from mothers to their offspring, and as a result, have evolved diverse strategies to increase the fitness of infected female hosts. There is also a great deal of applied interest in using various strains of Wolbachia to control insect pests and vectors of disease. 

This study takes advantage of the Darwin Tree of Life project, which aims to sequence the genomes of all eukaryotes in Britain and Ireland. Because Wolbachia is so pervasive, the researchers have been able to assemble and analyze over 100 high quality Wolbachia genomes, from symbionts infecting the first insects to have been sequenced in this project, mostly from Wyntham Woods near Oxford. This allows for the most comprehensive comparative genomic study to date of the 2 major Wolbachia supergroups (A and B) that infect insects, and a resource that will be heavily used by the Wolbachia and microbial symbiont community. There are many interesting findings, including clear demonstration that A and B group Wolbachia are distinct. A major question is how these two strains coexist in insects, and the authors find one interesting difference that may help answer this question, with A (but not) B group members containing an operon for arginine transport, suggesting that they take this nutrient from their hosts. Another interesting finding relates to biotin synthesis. Only a small fraction of Wolbachia harbor the biotin synthesis pathway, repeatedly acquiring this via horizontal gene transfer, and there has been speculation that providing biotin to hosts may be an important factor in establishment and persistence of some Wolbachia. Interestingly, the authors find that the biotin operon is strongly associated with toxin-antitoxin and selfish element genes, pointing to an interesting connection between selfish and 'mutualistic' Wolbachia genes and functions that merits further study.

In my opinion, this paper represents a very useful contribution to the Wolbachia and microbial symbiont field, and demonstrates the great potential of using high-quality eukaryote genome sequences to study their microbial infections. Also, the paper is beautifully written, and I found the figures to be of especially high quality. 

I don't have many comments or suggestions to strengthen the manuscript. Although the paper is very well-written and easy to read, I didn't find the discussion added very much new that wasn't already mentioned in the results or introduction. I would have also been interested to see more information about Wolbachia toxin evolution diversity, including some more phylogenies. The authors report the presence of spaid-like toxins, and it would be useful to get some more information about these. The spaid toxin was recently found to cause male-killing in Spiroplasma bacteria, so it would be interesting to understand what related genes are doing in Wolbachia (and how related they actually are). Finally, it would be interesting to learn more about the distribution and diversity of wmk genes, as these have been recently implicated in male-killing by Wolbachia. A recent study in biorxiv by Arai et al. found an interesting connection between male-killing and wmk copy number, and the high quality long-read sequence data presented in the current study has great potential to illuminate on this. 

Reviewer #3:

This manuscript reports 110 high quality new genome sequences of the bacterial symbiont Wolbachia. This resource is a major advance for the field of Wolbachia research. Not only does it roughly double the number of available genomes, but these genomes are of better quality and benefit from less biased sampling than those that are available already. There is a great deal of interest in the biology of this symbiont at the moment, with advances in understanding the genetics of its interaction with insect hosts and its deployment to control mosquito-borne viruses. As Wolbachia cannot be cultured outside of cells, research frequently relies on comparative studies. In the long term, the greatest contribution of this paper will likely be as a resource to this research community. 

As well as reporting the genomes themselves, the manuscript begins by describing the distribution of Wolbachia across host species and compares the host and symbiont phylogenies. There is already a substantial literature on these topics and this analysis does not lead to significant new insights. Nonetheless, having these patterns confirmed using whole-genome data is reassuring.

The remainder of the manuscript describes the properties of the genomes, and the richness of the data allows many new patterns or more complete patterns to be reported. For example, rearrangements within genomes, genetic exchange between genomes and gene content. The distribution of genes with a clear function, such a biotin synthesis, provides fascinating insights into the biology of Wolbachia. These results will be of great interest to the field.

I have a few minor suggestions that the authors may consider, but in my judgement none of them are necessary for publication.

Line 183-192. This is unsurprising given the literature on this topic. This data, like other datasets, shows that Wolbachia and host trees show many incongruences. However, it is also clear they are not independent. It seems that at the least this should be noted as a counterpoint to the observation of incongruences (eg a Mantel test comparing genetic distances of hosts and symbionts). In the future, this may be a nice resource to understand what predicts which species Wolbachia jumps between.

Figure 2C. This plot is most effective at showing synteny but it is used to demonstrate a recent host shift of Wolbachia between insect orders. There needs to be a clearer explanation of why this is the best way to show this - a tree seems more straightforward. I guess the answer is likely in Figure 3C, but this comes after.

Figure 3A. Is ANI just coding sequence? State In legend. The description of this in the text seems a bit odd as the correlation of sequence and structural divergence seems somewhat inevitable - maybe commenting on the degree of rearrangements might be more useful? It looks like synteny is pretty low except between very closely related strains.

Line 218-231. The definition of GC skew/skewI needs a clearer explanation. Explain why it is plotted against GC content. The interpretation of this statistic is a bit unclear. It is stated that groups A and B have low skew. Is this just relative to other supergroups, or bacteria in general? If the latter, does this translate directly into a measure of the rate of genome rearrangement.

Line 352-362. The distribution of different toxin genes is interesting but a bit hard to follow. A supplementary table or figure would make it more digestable. 

Line 406 'less likely' needs some justification. Line 412 seems to suggest ecological effects matter, and if there is preferential switching between hosts with shared ecology whis wil generate phylogenetic clustering. Its also unclear how 'host genetics' and 'wolbachia' genetics differ in this list - presumably you mean the interaction of the two.

Line 423-426. This seems like a very important pattern, but the text here does not seem to flow clearly though. 

Line 427. I guess you mean 'female hosts'. The rest of this paragraph could do with a few citations of similar work.

Are there any plasmids?

Fig S5. What is the statistical test?

---

## [Editor Report · Decision Letter 2]

19 Dec 2022

Dear Dr Vancaester,

Thank you for the submission of your revised Methods and Resources "Phylogenomic analysis of Wolbachia genomes from the Darwin Tree of Life biodiversity genomics project." for publication in PLOS Biology. On behalf of my colleagues and the Academic Editor, Luis Teixeira, I'm pleased to say that we can in principle accept your manuscript for publication, provided you address any remaining formatting and reporting issues. These will be detailed in an email you should receive within 2-3 business days from our colleagues in the journal operations team; no action is required from you until then. Please note that we will not be able to formally accept your manuscript and schedule it for publication until you have completed any requested changes.

Sincerely, 

Roli Roberts

Senior Editor

PLOS Biology

rroberts@plos.org